# Effect of a Topical Collagen Tripeptide on Antiaging and Inhibition of Glycation of the Skin: A Pilot Study

**DOI:** 10.3390/ijms23031101

**Published:** 2022-01-20

**Authors:** Young In Lee, Sang Gyu Lee, Inhee Jung, Jangmi Suk, Mun-Hoe Lee, Do-Un Kim, Ju Hee Lee

**Affiliations:** 1Department of Dermatology, Cutaneous Biology Research Institute, Yonsei University College of Medicine, Seoul 03722, Korea; ylee1124@yuhs.ac (Y.I.L.); dltkdrb5658@yuhs.ac (S.G.L.); 2Scar Laser and Plastic Surgery Center, Yonsei Cancer Hospital, Seoul 03722, Korea; 3Global Medical Research Center, Seoul 06526, Korea; ihjung@gmrc.co.kr (I.J.); rose@gmrc.co.kr (J.S.); 4Health Food Research and Development, NEWTREE Co., Ltd., Seoul 05604, Korea; mhlee@inewtree.com (M.-H.L.); dkim@inewtree.com (D.-U.K.)

**Keywords:** hydrolyzed collagen, tripeptide, glycation, advanced glycated end products, antiaging

## Abstract

The glycation process has been recognized as one of the critical parameters that accelerate signs of skin aging, especially in skin exposed to environment factors, such as ultraviolet radiation. Although previous studies showed the anti-inflammatory and antiaging properties of the hydrolyzed collagen tripeptide (CTP), its exact mechanism is not fully understood. Therefore, in this study, we sought to investigate the effect of a topical CTP on facial skin. Our group designed a 4 week prospective, single-arm study of 22 Asian women who applied topical CTP. We observed significant improvements in skin wrinkles, elasticity, and density with a reduction in skin accumulation of advanced glycated end products (AGEs) at week 4 without any adverse effects. The in vitro study revealed a preventive effect of the topical CTP on the accumulation of AGEs, denatured collagen production, and reactive oxygen species in dermal fibroblasts. Moreover, treatment with the CTP decreased induction of matrix metalloproteinases while increasing the collagen 1 level. These results suggest that the application of a topical CTP might improve clinical aging phenotypes via the inhibition of glycation and oxidative stress, leading to a delay in cellular aging.

## 1. Introduction

Skin aging is the combined result of the effects from intrinsic factors and external factors. In particular, external factors, such as ultraviolet radiation (UVR) exposure, have shown significant impacts on skin cell aging through DNA damage, increased oxidative stress, and the subsequent release of inflammatory mediators [1]. The consequent skin aging process results in aging skin phenotypes, such as wrinkles, irregular pigmentation, skin dryness, and decreased dermal and epidermal thickness [2]. As the average human life expectancy has increased, the general population worldwide has become increasingly interested in these skin aging outcomes, leading to the establishment of the aging-related nutraceutical market, which is one of the largest consumer markets today [3].

The hydrolyzed collagen tripeptide (CTP) is one of the popular ingredients considered to be an antioxidant that provides skin antiaging effects. The CTP is an enzymatically hydrolyzed form of collagen, enabling it to be used as a food supplement and for cosmeceutical skincare with better solubility and bioavailability along with lower allergenic properties compared to the traditional collagen formulations [4]. Previous studies on patients with atopic dermatitis (AD) and dry skin showed that oral and topical formulations of CTPs showed anti-inflammatory properties, resulting in improvement of AD, including skin hydration and itching [5]. Collagen hydrolysate has also been identified as a safe cosmetic ingredient for topical formulations with good moisturizing properties at the stratum corneum layer of the skin, thus reducing the effect of skin aging (dryness, laxity, and wrinkles) [6]. Although those previous studies showed the anti-inflammatory and antiaging properties of the CTP, its exact mechanism is not fully understood.

The aging process of skin is caused by DNA damage in nuclei and mitochondria, inflammation, glycation, decreased function of keratinocytes and fibroblasts, and the breakdown of heparan sulfate, hyaluronic acid, collagen, and elastin [7]. Among these multiple factors, glycation has been recognized as one of the critical parameters that accelerate signs of skin aging, especially in skin exposed to environmental factors, such as UVR. Glycation is the binding process of sugar to molecular structures, such as proteins, lipids, and nucleic acids, during which advanced glycated end products (AGEs) are formed. During the skin aging process, collagens, which are major components of the connective tissue that account for 30% of the human protein component, are modified by mineralization, the accumulation of AGEs, and depletion of glycosaminoglycans; these processes affect fiber stability and the skin’s susceptibility to matrix metalloproteinase (MMP)-mediated degradation [8].

AGEs, through the promotion of oxidative stress, activate several stress-induced transcription factors, with the production of proinflammatory mediators [9]. Moreover, AGEs alter the skin’s mechanical properties and, therefore, can be closely associated with the clinical manifestation of skin aging. Therefore, in this study, we aimed to investigate the effects of a topical CTP on antiaging of the skin via its antioxidant behavior and the inhibition of glycation process.

## 2. Results

### 2.1. Patient Characteristics

The clinical study included 22 Asian women with noticeable periorbital and glabellar wrinkles. Participants’ mean age was 47.1 years (range, 30–54 years). No participant dropped out because of noncompliance. In the final analysis after 4 weeks, data from 22 participants were collected and analyzed.

### 2.2. Clinical Efficacy and Safety of a Topical CTP on Skin Aging and AGE Accumulation

All participants applied the Ever Collagen Corrector Collagen Tripeptide Ampoule (CTP ampoule; NEWTREE Co., Ltd., Seoul, Korea), the commercialized cosmeceutical product used in the clinical study, twice daily onto their facial skin for 4 weeks. The total ingredient included in the test ampoule is listed in Appendix A. After 4 weeks, the mean periorbital skin roughness (Ra) measured by PRIMOS was significantly reduced from 20.77 ± 3.51 μm to 19.24 ± 3.52 μm (Figure 1A, *p* < 0.001). The mean glabellar Ra was also significantly reduced from 21.40 ± 3.09 μm to 20.21 ± 2.83 μm (Figure 1B, *p* < 0.001). In addition, the maximum periorbital wrinkle roughness (Rmax) significantly decreased from 191.11 ± 32.58 μm to 182.01 ± 33.63 μm (Figure 1C, *p* < 0.001). The maximum glabellar Rmax also significantly decreased from 169.92 ± 21.57 μm to 159.76 ± 21.58 μm (Figure 1D, *p* < 0.001).

Meanwhile, clinical improvement of wrinkles was also visualized by a three-dimensional (3D) skin analysis camera system (Antera 3D^®^, Miravex, Dublin, Ireland). The mean depressed depth of periorbital wrinkles at baseline was 0.06 ± 0.01 mm, which significantly improved to 0.05 ± 0.01 mm at week 4 (Figure 2A,B, *p* < 0.001). 

Within 4 weeks, the skin density measured by Ultrascan UC22 (Courage + Khazaka electronic GmbH, Köhn, Germany) showed a significant increase from 55.66 ± 7.61 to 59.67 ± 7.84 (Figure 3A,B, *p* < 0.001). Furthermore, the skin surface elasticity (total recovery/total elongation; gross elasticity (R2)) increased from 0.81 ± 0.03 to 0.83 ± 0.03 (Figure 3C, ** p* < 0.001). The maximum collagen strength (R4) after application of the topical CTP also significantly increased from 68.02 ± 5.48 to 70.24 ± 5.14 (Figure 3D, *p* < 0.001). Meanwhile, AGE accumulation in the skin was 2.26 ± 0.32 AU at baseline, and it was significantly reduced to 2.16 ± 0.29 AU at week 4 (Figure 3E, *p* < 0.001). The summary of the overall clinical studies data outcomes is listed in Appendix A. 

Overall, the average percentage of subjects who responded to “very satisfied” and “satisfied” to the 4 week use of topical CTP product and the overall improvement in facial skin problems caused by aging at the end of the study was 95.5% (“very satisfied”: four subjects, “satisfied”: 17 subjects). One subject rated “slightly satisfied” with respect to the test product and the overall improvement of the facial skin. No adverse events were observed during the 4 week study period, and none of the participants dropped out of the study because of adverse events, suggesting that the test product formulation was safe to use.

### 2.3. In Vitro Evaluation of the Effect of EverCTP^TM^ on Cellular Aging

As the cell viability of 1000 μg/mL of the hydrolyzed fish skin extract (EverCTP^TM^; NEWTREE Co., Ltd., Seoul, Korea) was 96.88%, the investigators concluded that EverCTP^TM^ had no cell cytotoxicity with treatment up to 1000 μg/mL (Appendix A).

To investigate the effect of EverCTP^TM^ on cellular aging, we performed a quantitative real-time reverse transcription polymerase chain reaction (qRT-PCR) to measure collagen 1, matrix metallopeptidase (MMP)1, *MMP3*, and *MMP9* messenger (m)RNA expression levels. UVR exposure was used in this experiment for the generation of extrinsic cellular aging on HDFs. Transforming growth factor-beta (TGF-β; Peprotech, Rocky Hill, NJ, USA) and retinoic acid (RA; Sigma Aldrich, Saint Louis, MO, USA) were used as positive controls for collagen 1 and MMPs, respectively. When compared to the control group, cells treated with 500 μg/mL and 1000 μg/mL of EverCTP^TM^ showed significantly higher expressions of collagen 1 (Figure 4A, *p* < 0.05 and *p* < 0.005, respectively). The expression levels of MMP-1, -3, and -9 reduced significantly after the treatment with 250 μg/mL, 500 μg/mL, and 1000 μg/mL of EverCTP^TM^ compared to the UVR exposure group. (Figure 4B–D; *p* < 0.05, *p* < 0.01, and *p* < 0.005, respectively).

The procollagen type I peptide (PIP1) concentration increased significantly when participants were treated with 250 μg/mL and 500 μg/mL of EverCTP^TM^ compared to the control (Figure 5A, *p* < 0.01 and *p* < 0.005, respectively). Treatment with 1000 μg/mL of EverCTP^TM^ also increased the PIP1 concentration (885.65 ± 89.69) compared to the control (824.65 ± 8.12), but this was not statistically significant. On the other hand, the MMP1 protein secretion after ultraviolet (UV) exposure showed a significant reduction when treated with 250 μg/mL, 500 μg/mL, and 1000 μg/mL of EverCTP^TM^ (Figure 5B, *p* < 0.005).

To measure the elastase activation that increases due to UV light and reactive oxygen species (ROS) during the aging process, an elastase inhibition assay was performed. Accordingly, 1 mM phosphoramidon (PR; Sigma Aldrich) was used as the positive control. Elastase inhibition activity was significantly increased after treatment with EverCTP^TM^ dose-dependently. Compared to the negative control, 250 μg/mL, 500 μg/mL, and 1000 μg/mL of EverCTP^TM^ showed 42.12%, 42.74%, and 48.47% increases in elastase inhibition activity, respectively (Figure 6, *p* < 0.005).

### 2.4. In Vitro Evaluation of the Effect of EverCTP^TM^ on Reactive Oxygen Species and the Glycation Process

After treatment with 10 μM hydrogen peroxide (H_2_O_2_), the increased reactive oxygen species (ROS) level was significantly reduced when cotreated with 1000 μg/mL of EverCTP^TM^ (Figure 7, *p* < 0.05, *p* < 0.01, *p* < 0.005). Meanwhile, the effect of EverCTP^TM^ on the glycation process in human dermal fibroblast (HDF) cells was investigated by enzyme-linked immunosorbent assay (ELISA) after UV exposure. Treatment with 1000 μg/mL of EverCTP^TM^ significantly decreased the productions of pentosidine (from 211.682 ± 0.425 to 156.975 ± 0.135), methylglyoxal (from 5.586 ± 0.014 to 5.307 ± 0.001), and AGEs (from 16.688 ± 0.235 to 10.762 ± 0.424), while significantly increasing the production of glyoxalase I compared to the UV group (from 0.008 ± 0.001 to 0.777 ± 0.003; Figure 8, *p* < 0.005).

Additional investigations of the CTP ampoule were performed to assess its effect on the productions of glycated collagen and AGEs. The fluorescence intensity of glycated collagen production of the control group was 0.999 ± 0.058, while that of the CTP ampoule control group without collagen component (CTP control) was 0.09 ± 0.027 (Figure 9A, *p* < 0.005). Compared to the control group, the CTP ampoule group (with collagen) showed a significant decrease in glycated collagen production to 0.517 ± 0.114 (Figure 9A, *p* < 0.005). Moreover, the fluorescence intensity of AGE production of the control group was 2.476 ± 0.005, while that of CTP ampoule group was significantly decreased to 2.296 ± 0.002 (Figure 9B, *p* < 0.005). Lastly, we measured the degree of denatured collagen production induced by heat treatment to observe the effect of CTP ampoule cotreatment. The fluorescence intensity of denatured collagen production after heat treatment was 13.361 ± 2.532, and the degree of intensity significantly decreased to 8.676 ± 0.623 after treatment with the CTP ampoule (Figure 9C, *p* < 0.05).

### 2.5. Assessment of Skin Absorption after Topical Application of EverCTP^TM^

The efficacy of collagen 1 absorption after topical application of gelatin (nonhydrolyzed collagen) and EverCTP*^TM^* to the skin was evaluated in reconstructed human micro-tissue (KeraSkin-FT, Bio Solution Co., Ltd., Seoul, Korea). The fluorescence intensity of collagen 1 absorbed into the skin after 1000 μg/mL of EverCTP^TM^ treatment was 2.316 ± 0.444, which was increased compared to that after 1000 μg/mL of gelatin treatment (2.051 ± 0.550) and the negative control (0.266 ± 0.111; Figure 10A,B, *p* < 0.005). Despite an increase in the EverCTP^TM^ treatment group, the difference in florescent intensities compared to the gelatin group did not reach a statistical significance (*p* = 0.5520). 

Additionally, the results of the additional quantitative evaluation on the rates of collagen absorption via high-performance liquid chromatography (HPLC) compared between the gelatin and EverCTP^TM^ treatments are shown in Figure 10C. The rates of collagen absorption in the gelatin treatment group and EverCTP^TM^ treatment group were 0.425 ± 0.100 and 12.221 ± 0.684, respectively, thus showing a significantly improved quantitative absorption rate of collagen in the EverCTP^TM^ group (*p* < 0.005).

## 3. Discussion

Recently, both topical and oral collagen-containing products have become popular for their antiaging properties [3]. Although the antioxidant ability of the CTP is due to the presence of hydrophobic amino acids in the peptide [10], the exact molecular mechanism underlying how the CTP acts as an antiaging product has not been fully uncovered. A previous placebo-controlled clinical trial demonstrated that oral ingestion of the CTP helped retain skin water contents [11]. Moreover, a recent study on the cutaneous hydration effect of CTP intake in mice showed significant reductions in transepidermal water loss, scratching behavior, and levels of hyaluronidase-1, tumor necrosis factor-alpha, and interleukin-6, while showing increased water content and hyaluronan synthase-2 levels; these results suggested that CTP products could enhance skin hydration with potential as a skin hydration agent [12]. The present study showed improvements in facial wrinkles, skin elasticity, and density without any adverse effects after use of the topical CTP. The participants’ satisfaction scores also subjectively showed an overall improvement of their facial skin. Moreover, there was a significant reduction in AGE autofluorescence at week 4, implying that the topical CTP has both antiaging and anti-glycation effects clinically.

To further investigate the molecular mechanism underlying the antiaging effect of the topical CTP, we performed an in vitro study on HDF cells. After UV exposure of fibroblasts to induce cellular senescence and aging, expressions of *Collagen1*, *MMP1*, *MMP3*, and *MMP9* mRNA with or without CTP treatment were measured. The *Collagen1* gene expression significantly increased with CTP treatment, whereas *MMP1*, *MMP3*, and *MMP9* expressions significantly decreased dose-dependently. Moreover, ELISA of PIP1 and MMP1 levels in UV-exposed HDF cells with CTP treatment showed an increased PIP concentration, while MMP1 levels decreased with CTP treatment, suggesting that the CTP showed an antiaging effect on fibroblasts. The additional elastase inhibition assay showed significantly increased elastase inhabitation activity after CTP treatment, further indicating its antiwrinkle effect.

Over recent years, various benefits of supplementation with the CTP have been reported including improvements in joint pain, wound healing, blood pressure, glucose tolerance, elasticity and wrinkles of the skin, and epidermal barrier function [4,13,14]. Moreover, the CTP has been shown to stimulate cell proliferation in dermal fibroblasts, while increasing synthesis of hyaluronic acid and acceleration of cell migration [15,16,17]. During the process of skin aging, a profound atrophy of dermal connective tissue including collagen occurs, which impairs strength and resiliency of the skin. Glycation and production of AGEs further debilitate the function of collagen, making it stiffer and more brittle [17]. The glycation process also leads to profound changes in the behavior of dermal fibroblasts, reducing their proliferation and migration, while simultaneously disrupting collagen I maturation and preventing collagen deposition in the extracellular matrix [18].

Herein, we investigated the relationship between the application of a CTP and reductions in the glycation process. Previous studies have shown that the production of AGEs results in the activation of nuclear factor-kappa beta (NF-κB) transcription factors via the generation of oxygen radicals and MAP kinase signaling; the NF-κB activation subsequently leads to the induction of MMPs and formation of various proinflammatory cytokines, which leads to cellular aging [19]. Our in vitro study revealed that the ROS expression level of H_2_O_2_-treated HDF cells was significantly reduced when cotreated with the CTP, implying its antioxidant effect. Furthermore, the subsequent ELISA with UV-exposed HDF cells showed a significant decrease in productions of pentosidine and methylglyoxal, the two major AGE biomarkers, after treatment with the CTP. However, the level of glyoxalase-1, a ubiquitous cellular enzyme that participates in the detoxification of methylglyoxal, significantly increased upon treatment with the CTP. The in vitro study of the CTP ampoule also showed significant reductions in the production of glycated collagen, AGEs, and denatured collagen in the treatment group compared to the control group. These results suggest that supplementation with the CTP not only provided an antioxidant effect, but also inhibited AGE accumulation, which could lead to the reduction in MMPs and proinflammatory cytokines, thus promoting overall antiaging of the skin.

The antioxidant properties of the CTP depend on the lower molecular weight of peptides, as smaller molecules with average molecular weights of 5 kDa showed greater ability to donate an electron or hydrogen to stabilize oxygen radicals than larger molecules [20]. Since the absorption capacity of a topical collagen product through the skin barrier is frequently limited by its molecular size, manufacturing small (1–10 kDa) collagen hydrolysate with excellent biocompatibility, easy biodegradability, and weak antigenicity is crucial for utilization [6,21]. In our study, we compared the absorption rate of EverCTP^TM^, with gelatin—a form of a heterogeneous mixture of peptides derived from the parent protein collagen [21]. The EverCTP^TM^ group showed a significantly improved collagen absorption rate compared to the gelatin group, suggesting the enhanced biocompatibility of the CTP formulation. 

In conclusion, our pilot study showed the effect of a topical CTP formulation on antiaging and inhibition of the glycation process of skin. These results suggest that the application of a topical CTP might improve clinical aging phenotypes via the inhibition of glycation and oxidative stress, leading to a delay in cellular aging. Nevertheless, our study had several limitations. First, it included a small number of samples and short follow-up period in the clinical study. Hence, a further clinical study with a larger sample size and longer follow-up period with a double-blinded, controlled design is required. Secondly, an additional skin absorption study on collagen 1 between EverCTP^TM^ and the simple tripeptide, Gly–Pro–Hyp, would further show efficacy and biocompatibility of the test product in its topical formulation. Lastly, although our in vitro study revealed not only a reduction in ROS and AGE production but also an increase in collagen 1 and the reciprocal reduction of MMPs, additional studies on the molecular pathogenesis are needed to provide a complete understanding of the effect of a topical CTP on antiaging of the skin. 

## 4. Materials and Methods

### 4.1. Study Design and Ethical Considerations

This prospective, single-arm clinical trial was approved by the Institutional Review Board (IRB) of the Global Medical Research Center (IRB number: GIRB-21624-DX), and informed consent was obtained from each participant. The investigation was performed in full compliance with the principles of Good Clinical Practices and the Declaration of Helsinki.

All participants were instructed to apply the test product, the CTP ampoule, to their entire face twice daily for 4 weeks to evaluate its clinical efficacy and safety. To assess its safety, all participants were asked to report any adverse events they experienced while using the topical product.

### 4.2. Participants

Twenty-two healthy women aged between 30 and 50 years with noticeable periorbital and glabellar wrinkles were considered eligible for inclusion in the study. The main exclusion criteria were as follows: major internal disorders, major or active skin diseases, current pregnancy or lactation, history of allergy or hypersensitivity reactions, history of undergoing dermatologic procedures (e.g., lasers, filler, botulinum toxin injection) in the past 3 months, use of topical medications with any skin reactions (e.g., glucocorticoids, retinoids, and topical immunomodulators) in the past 3 months, use of systemic medications, such as glucocorticoids and immunomodulators within the last 1 month, and application of skin care products or topical medications with similar functions to topical collagen peptides within the last 3 months before participating in the study.

### 4.3. Test Product

The CTP ampoule containing a hydrolyzed fish skin extract (EverCTP^TM^) was used in the clinical study. EverCTP^TM^ is a form of collagen hydrolysate obtained from the skin of *Pangasius hypophthalmus* that contains 4% Gly–Pro–Hyp with the tripeptide content exceeding 25%; it was used as the test ingredient in our laboratory study. The preliminary human skin ex vivo penetration study with analysis of Franz diffusion cells showed 6.74% permeability of the Gly–Pro–Hyp tripeptide.

### 4.4. Clinical Efficacy Assessment

All participants were followed up at baseline and week 4 after the initiation of the treatment. The clinical efficacy of EverCTP^TM^ was investigated by assessing biophysical parameters determined by an optical 3D measurement system (PRIMOS CR, SnT Lab, Seoul, Korea), Cutometer Dual MPA580 (Courage Khazaka electronic GmbH, Köln, Germany), and Ultrascan UC22 (Courage + Khazaka electronic GmbH). The wrinkle volume and skin roughness of the periorbital area (crow’s feet) and glabellar area were measured by the phase-shift rapid in-vivo measurement of skin (PRIMOS) at baseline and week 4. Among the skin surface descriptors of PRIMOS, Ra and Rmax were chosen to measure the 3D wrinkle changes during the study period. Antera 3D was used to perform a quantitative assessment of the depressions caused by the wrinkles.

Changes in skin elasticity were measured using the Cutometer MPA580. Among the various parameters of the Cutometer, we chose R2 and R4 to represent the efficacy of the test product in improvement of skin elasticity. The Ultrascan UC22 was used to measure changes in the density of the skin.

To noninvasively assess AGE accumulation in the skin, we used skin autofluorescence (skin AF) measured by the AGE Reader (DiagnOptics Technologies B.V., Groningen, The Netherlands). Skin AF, expressed as arbitrary units (AU), was measured in triplicate at the skin site on participants’ inner forearm, which is a body part easily accessible and rarely exposed to the sun, as previously described [22]. 

All subjects underwent a physical examination to assess safety outcomes at every visit. Participants were also asked to report side-effects during the study period. Additionally, the participants were questioned to score their rates of satisfaction on the use of topical CTP product and the overall improvement in facial skin problems caused by aging using the following scale: 1 = unsatisfied, 2 = no change, 3 = slightly satisfied, 4 = satisfied, and 5 = very satisfied. 

### 4.5. Cell Culture

HDF cells purchased from ATCC^®^ (Manassas, VA, USA) were cultured in Dulbecco’s modified Eagle’s medium (Lonza, Walkersville, MD, USA) supplemented with 1% penicillin–streptomycin (Gibco, Grand Island, NY, USA) and 10% fetal bovine serum (Gibco). HDF cells were incubated in a humidified atmosphere containing 5% carbon dioxide (CO_2_) at 37 °C. TGF-β, RA, and PR were used as appropriate positive controls. UV exposure and the treatment with H_2_O_2_ were performed for the generation of extrinsic cellular aging and the induction of ROS stress on HDFs, respectively. 

### 4.6. Cell Viability Measurement

To determine if EverCTP^TM^ was cytotoxic, a cell counting kit-8 (CCK-8; Dojindo, Mashiki, Japan) assay was used according to the manufacturer’s instructions to evaluate HDF cells (ATCC^®^). In brief, HDF cells (5 × 10^4^ cells/well) were seeded onto a 96-well plate. When cell confluence was >80%, the HDF cells were treated with EverCTP^TM^ and incubated for 24 h. The ELISA microplate reader (VersaMax; Molecular Devices, San Jose, CA, USA) was used to measure the absorbance at 450 nm.

### 4.7. Total RNA Extraction and qRT-PCR

Total RNA was extracted from the HDF cells treated with EverCTP^TM^ after no irradiation or 10 mJ/cm^2^ of ultraviolet-B (UVB) irradiation using a UVB lamp (BLX26, BIO-LINK^®^-Crosslinker, FRA). Using the TRIzol reagent (Invitrogen, Waltham, MA, USA) according to the manufacturer’s instructions, total RNA extraction was performed. For quantification and reverse transcription of total RNA, the NanoDrop 2000 spectrophotometer (ThermoFisher Scientific, Carlsbad, CA, USA) and RNA to complementary DNA EcoDry Premix Kit (Takara Sake, Berkley, CA, USA) were used. qRT-qPCR was conducted using the SYBR Green Master MIX (4309155, Promega Co., Madison, WI, USA), specific primer pairs, and the QuantStudio 3 Real-Time Polymerase Chain Reaction System (Applied Biosystems, Waltham, MA, USA). Primer sequences used in the present study are shown in Appendix A. Relative mRNA expression levels were normalized to *GAPDH* and calculated using the 2^−ΔΔCt^ method.

### 4.8. Enzyme-Linked Immunosorbent Assay

PIP and MMP1 protein concentrations were detected using ELISA kits (Takara, Kusatsu, JPN; Amersham Life Science, Amersham, UK). MMP1 production was induced by 10 mJ/cm^2^ of UVB irradiation. The Human Pentosidine ELISA kit (MyBioSource, San Diego, CA, USA), Methylglyoxal ELISA kit (Abcam, Cambridge, UK), Glyoxalase I assay kit (Abcam), and AGE assay Kit (Abcam) were used to measure pentosidine, methylglyoxal, gyloxalase I, and overall AGE production. All kits were used according to the manufacturers’ instructions. The absorbance detection was performed using an ELISA microplate reader (VersaMax) at 450 nm.

### 4.9. Elastase Inhibitory Assay

To measure elastase inhibition activity, an elastase solution extracted by HDF cells was prepared by freezing/thawing the cells more than three times. After the solution was centrifuged (3000 rpm) at 4 °C for 20 min, the supernatant was harvested and used as the elastase solution. The elastase solution was placed in 100 μg of protein per well on a 96-well plate, and we added 0.2 M Tris–HCl buffer (pH 8.0) to make the total volume 88 μL. We added 2 μL of *N*-succinyl-tri-alanyl-*p*-nitroanilide, a substrate of elastase, and 10 μL of EverCTP^TM^ of three concentrations or 10 μL of 1 mM PR (Sigma Aldrich, Saint Louis, MO, USA) to each well containing the elastase solution and cultured the plate at 37 °C for 90 min. We added 10 μL of 0.2 M Tris–HCl buffer as a control group. Finally, using an ELISA microplate reader (VersaMax), the absorbance was measured at 405 nm at 37 °C. The elastase inhibition activity was expressed as a decrease (%) in absorbance of the test group with and without the sample solution.
Elastase inhibition activity (%) = (A − B)/A × 100,
where A is the absorbance of control, and B is the absorbance of sample.

### 4.10. ROS Assessment

Intracellular ROS accumulation assessment was performed using the 2′7′-dichlorofluorescin diacetate (DCFDA) cellular ROS Detection Assay Kit (Abcam). Briefly, the HDF cells (5 × 10^4^ cells/dish) were seeded onto a glass bottom dish, and, when the cell confluence was >80%, we added 200 μL of 25 μM DCFDA to each dish and incubated the cells at 37 °C for 45 min. To induce ROS stress as the positive control, the treatment with 10 μM H_2_O_2_ was performed. For the test group, 10 μM H_2_O_2_ was added and incubated for 24 h prior to DCFDA. After diffusion into the cells, DCFDA was deacetylated and subsequently oxidized by ROS to form dichlorofluorescein, which is highly fluorescent, and it was assessed by confocal microscopy. After discarding the DCFDA solution, each dish was washed using phosphate-buffered saline (PBS). Fluorescence intensity was imaged and measured using a confocal microscope (LSM 700, Carl Zeiss, Jena, Germany), and compared between groups using ImageJ software (National Institutes of Health, Bethesda, MD, USA).

### 4.11. Assessment of Glycated Collagen Production and AGEs

Glycated collagen production was estimated using the Collagen Glycation Assay Kit (COSMO BIO, Tokyo, Japan), which was used according to the manufacturer’s instructions. In brief, 50 μL of a collagen solution was placed onto a 96-well black plate and incubated at 37 °C for 18 h. After it reacted for 18 h, the sample dilution buffer was added to the buffer group. In order to check the glycation reaction of the test product itself, the topical CTP ampoule without collagen was added to the CTP control group (without collagen). In the control group, fructose was added, while both fructose and the topical CTP ampoule (with collagen) were added to the CTP ampoule group. Using the microplate reader (VersaMax) at 370 nm and 440 nm, the absorbance was immediately detected after 4 weeks of reaction at 37 °C. Detected absorbance was used to compare the glycated collagen production between the groups.

In the assessment of AGE production, a mixture of bovine serum albumin (BSA) (5 mg/mL; Sigma-Aldrich, St. Louis, MO, USA), glucose (25 mM), and PBS or the CTP ampoule was placed in a 1:1:1 ratio and reacted at 37 °C for 2 weeks. PBS only was added to the buffer group, while BSA and glucose were added to the control group. The CTP ampoule was additionally treated along with BSA and glucose in CTP ampoule group. In order to verify the glycation reaction of the CTP ampoule itself, a control for the CTP ampoule excluding BSA was tested as the CTP control group. 

### 4.12. Immunofluorescence Staining for Evaluating Denatured Collagen Production

The HDF cells (6 × 10^4^ cells/well) were seeded on four-well chamber slides and incubated with the CTP ampoule for 24 h when cell confluence reached > 80%. The slides were washed, treated with heated PBS (95 °C) for 1 min, and fixed with 4% paraformaldehyde. To induce denatured collagen production by HDFs as the positive control, heat stimulation was performed. After fixation, the slides were incubated with a denatured collagen detection reagent (Funakoshi, Tokyo, Japan) and incubated with FITC-conjugated streptavidin. Finally, the slides were fixed using the VECTASHIELD^®^ antifade mounting medium with DAPI (Vector Laboratories Inc., Burlingame, CA, USA) and evaluated using a fluorescence microscope (M2, Carl Zeiss). The fluorescence intensity was calculated using obtained fluorescence images and ImageJ software (National Institutes of Health).

### 4.13. Immunofluorescence Staining and HPLC to Assess Collagen 1 Absorption

The reconstructed human micro skin tissue model KeraSkin-FT was cultured in the KeraSkin-FT growth medium (Bio Solution Co., Ltd.) and incubated for 24 h in a humidified atmosphere containing 5% CO_2_ at 37 °C after covering with 10 μL of 1000 μg/mL of EverCTP^TM^ or gelatin (non-hydrolyzed collagen) extracted from fish skin. Following fixation with 4% paraformaldehyde, the samples were made into 6 µm thick frozen sections. The frozen sections were incubated with primary anti-fish collagen 1 (Abcam) and a goat anti-rabbit immunoglobulin G H&L secondary antibody (Abcam). Then, the frozen sections were fixed using the VECTASHIELD^®^ mounting medium with DAPI and evaluated using a fluorescence microscope. The fluorescence intensity was calculated using obtained fluorescence images and ImageJ software (National Institutes of Health).

Additionally, after incubation, the medium containing collagen absorbed through the reconstructed human micro skin tissue model was analyzed to quantify hydroxyproline using HPLC. The increased absorption rates were calculated as previously described.

### 4.14. Statistical Analyses

All data were expressed as the mean ± standard deviation. Datasets were assessed for normality using the Kolmogorov–Smirnov test. Clinical parameters at baseline and week 4 were compared between groups using the Wilcoxon signed-rank test or paired samples *t*-test according to the results from the normality test. The data were statistically analyzed using SPSS statistics version 25.0 software (IBM Corp., Armonk, NY, USA). Mean differences were considered significant when *p* < 0.05, *p* < 0.01, and *p* < 0.005. All laboratory experiments in the present study were performed at least three times (*n* ≥ 3); for statistical data analyses, the obtained data were used.

## Figures and Tables

**Figure 1 ijms-23-01101-f001:**
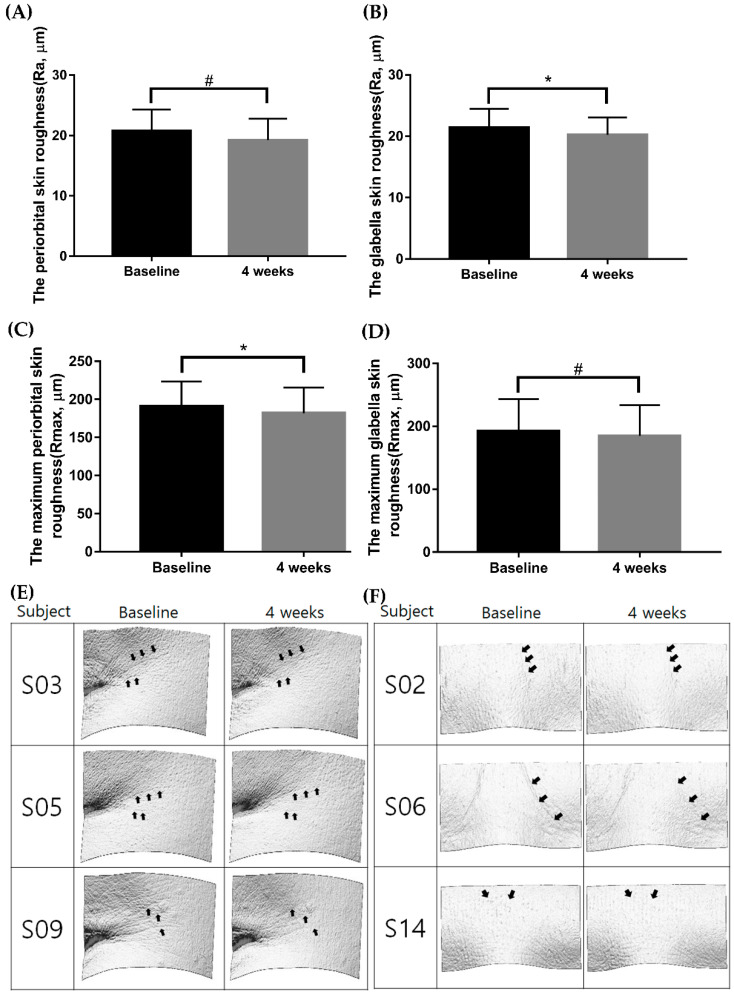
Comparison of skin roughness and maximum skin roughness before and after 4 weeks of applying the CTP ampoule. Skin roughness and maximum skin roughness were reduced in response to application of the CTP ampoule over 4 weeks (**A**–**D**); # *p* < 0.001, Wilcoxon signed-rank test; * *p* < 0.001, paired samples *t*-test). The 4 week follow-up images taken with PRIMOS (PRIMOS CR, SnT Lab, Seoul, Korea) are shown in (**E**,**F**). CTP ampoule, Ever Collagen Corrector Collagen Tripeptide Ampoule.

**Figure 2 ijms-23-01101-f002:**
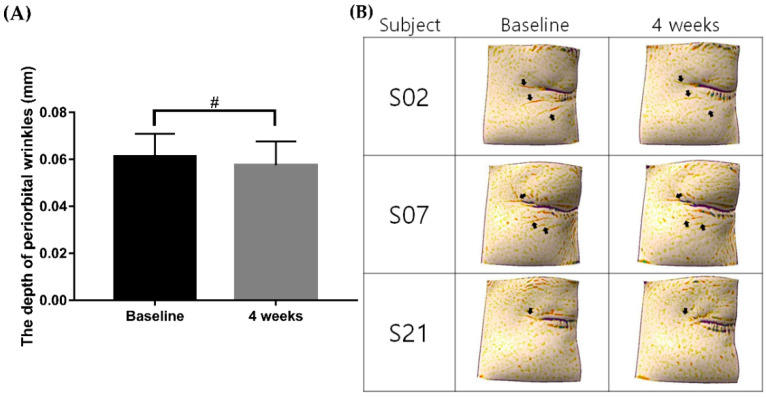
Comparison of the depth of periorbital wrinkles before and after 4 weeks of applying the CTP ampoule. The depth of periorbital wrinkles was reduced in response to application of the CTP ampoule over 4 weeks (**A**); # *p* < 0.001, Wilcoxon signed-rank test). The 4 week follow-up images taken with the Antera 3D^®^ (Miravex, Dublin, Ireland) are shown in (**B**). CTP ampoule, Ever Collagen Corrector Collagen Tripeptide Ampoule.

**Figure 3 ijms-23-01101-f003:**
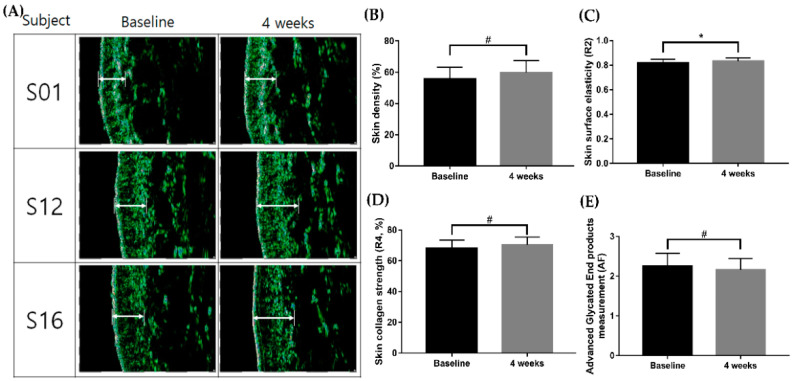
Comparison of skin density, skin surface elasticity, skin collagen strength, and the AGE measurement before and after 4 weeks of applying the CTP ampoule. Skin density, skin surface elasticity, skin collagen strength, and the AGE measurement were increased in response to applying the CTP ampoule over 4 weeks (**B**–**E**); # *p* < 0.001, Wilcoxon signed-rank test; * *p* < 0.001, paired samples *t*-test. The 4 week follow-up images taken with the Ultrascan UC22 (Courage + Khazaka electronic GmbH, Köhn, Germany) are shown in (**A**). CTP ampoule, Ever Collagen Corrector Collagen Tripeptide Ampoule; AGE, advanced glycated end product.

**Figure 4 ijms-23-01101-f004:**
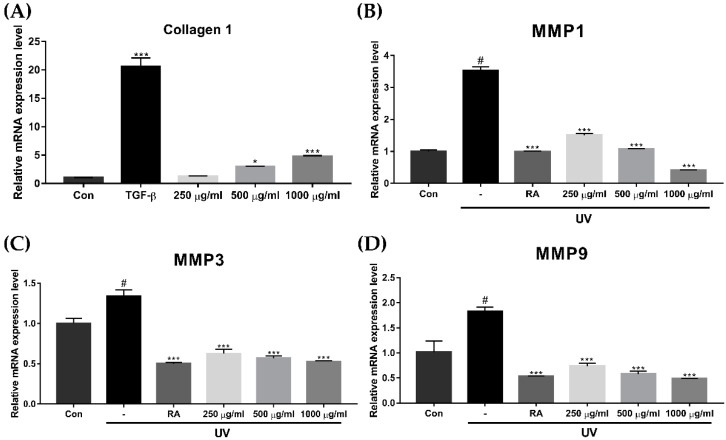
Relative messenger RNA expression levels of *Collagen 1*, *MMP1*, *MMP3*, and *MMP9* in the HDF cells. The collagen 1 gene expression level was increased concentration dependently in the EverCTP^TM^ (NEWTREE Co., Ltd., Seoul, Korea) treatment group compared to the CON group (**A**). *MMP1* (**B**), *MMP3* (**C**), and *MMP9* (**D**) gene expressions were induced by 10 mJ/cm^2^ of UVB irradiation, and induced MMP family gene expressions were decreased with EverCTP^TM^ treatment; # *p* < 0.005, independent samples *t*-test compared with the CON group; * *p* < 0.05, *** *p* < 0.005, independent samples *t*-test compared with the CON group (**A**) or the UV irradiation group (**B**–**D**). TGF-β, transforming growth factor-beta; RA, retinoic acid; MMP, matrix metalloproteinase; HDF, human dermal fibroblast; CON, control; UVB, ultraviolet-B; UV, ultraviolet.

**Figure 5 ijms-23-01101-f005:**
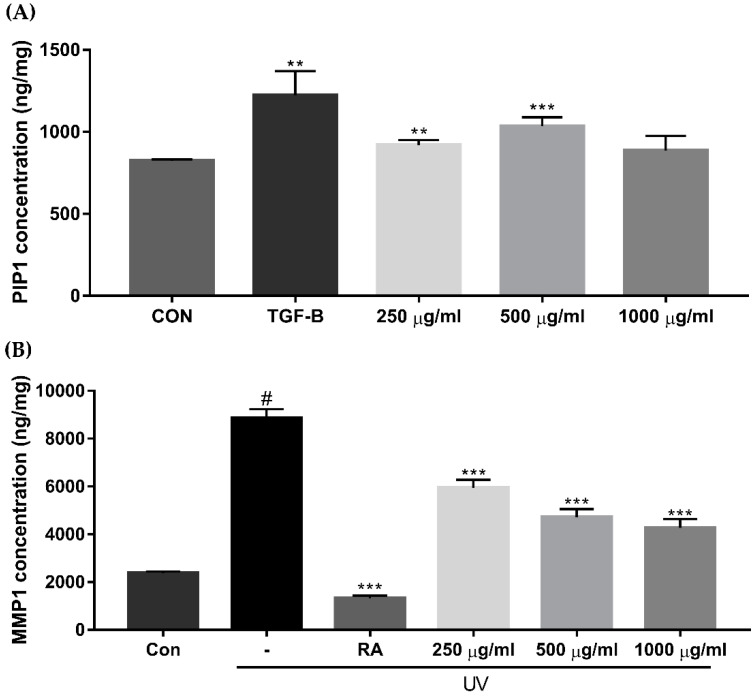
Concentrations of PIP1 and MMP1 in HDF cells according to the ELISA assay. The PIP1 concentration was significantly increased after treatment with 250 μg/mL and 500 μg/mL of EverCTP^TM^ (NEWTREE Co., Ltd., Seoul, Korea) and TGF-β compared to the CON (**A**). MMP1 production was induced by 10 mJ/cm^2^ of UVB irradiation, and induced MMP1 production was dose-dependently decreased according to the concentration of EverCTP^TM^ treatment (**B**); # *p* < 0.005, independent samples *t*-test compared with the CON; ** *p* < 0.01, *** *p* < 0.005, independent samples *t*-test compared with the CON group (**A**) or the UV irradiation group (**B**). TGF-β and RA were used as the positive controls. TGF-β, transforming growth factor-beta; RA, retinoic acid; PIP1, procollagen type I; MMP, matrix metalloproteinase; ELISA, enzyme-linked immunosorbent assay; CON, control; UVB, ultraviolet-B; UV, ultraviolet; HDF, human dermal fibroblast.

**Figure 6 ijms-23-01101-f006:**
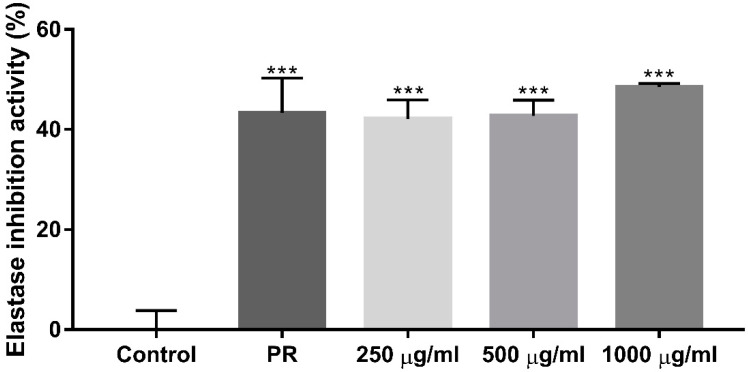
Percentage of elastase inhibition activity according to PR (positive control) and 250, 500, and 1000 μg/mL of EverCTP^TM^ (NEWTREE Co., Ltd., Seoul, Korea) treatment. The treatment increased elastase inhibition activity as much as PR. *** *p* < 0.005, independent samples *t*-test compared with the control. PR, phosphoramidon.

**Figure 7 ijms-23-01101-f007:**
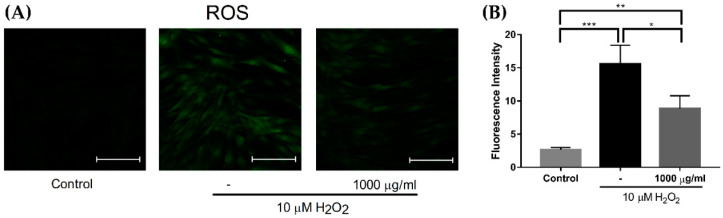
Visualization of ROS production in HDF cells (**A**). ROS production was stimulated by 10 μM of H_2_O_2_ treatment and significantly reduced by 1000 μg/mL of EverCTP^TM^ (NEWTREE Co., Ltd., Seoul, Korea) treatment (**B**); * *p* < 0.05, ** *p* < 0.01, *** *p* < 0.005, independent samples *t*-test. Scale bar indicates 100 μm. ROS, reactive oxygen species; HDF, human dermal fibroblast; H_2_O_2_, hydrogen peroxide.

**Figure 8 ijms-23-01101-f008:**
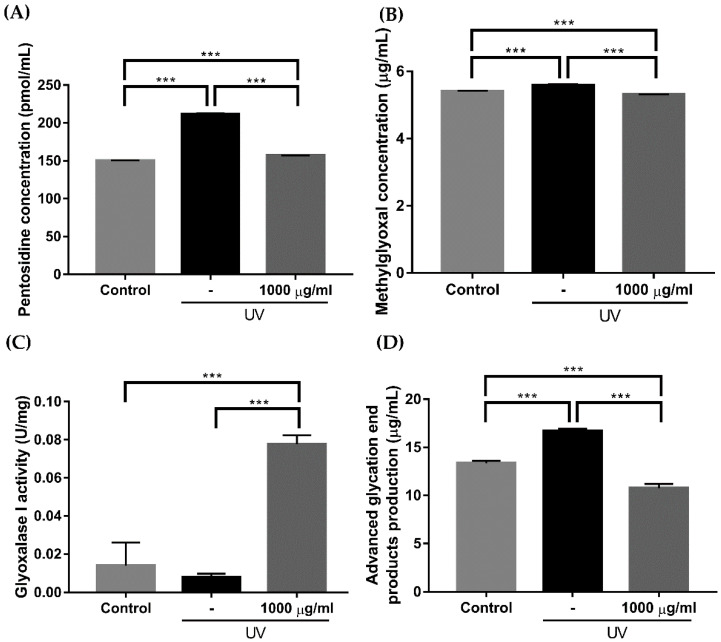
Comparison of the pentosidine, methylglyoxal, glyoxalase 1 activity, and advanced glycation end product (AGE) production in HDF cells before and after the treatment. Pentosidine, methylglyoxal, and AGE production was induced by 10 mJ/cm^2^ of UVB irradiation (**A**,**B**,**D**). The induced pentosidine, methylglyoxal, and AGE concentrations were significantly reduced after treatment with 1000 μg/mL of EverCTP^TM^ (NEWTREE Co., Ltd., Seoul, Korea). Glyoxalase 1 activity was reduced by 10 mJ/cm^2^ of UVB irradiation, and the reduced glyoxalase 1 activity was significantly increased with treatment of 1000 μg/mL of EverCTP^TM^ (**C**); *** *p* < 0.005, independent samples *t*-test. HDF, human dermal fibroblast; ultraviolet-B; UV, ultraviolet.

**Figure 9 ijms-23-01101-f009:**
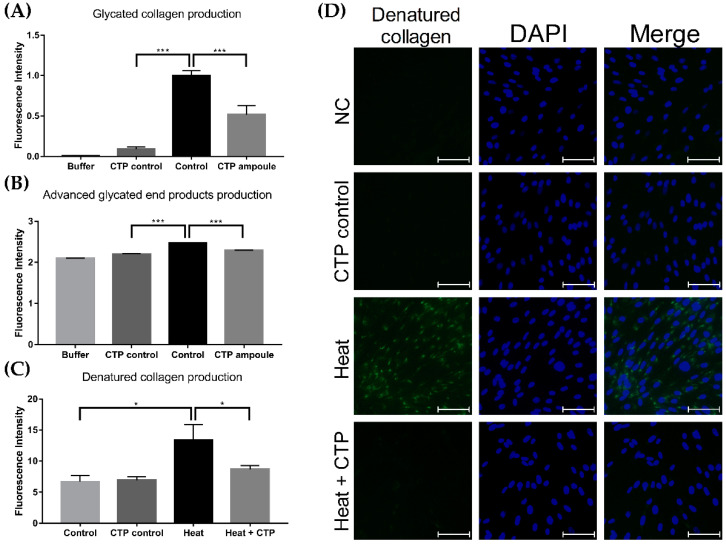
Effect of the CTP ampoule on glycated collagen, AGE, and denatured collagen production. Glycated collagen and AGE production decreased in the CTP ampoule treatment group compared to the control group (**A**,**B**). Denatured collagen production induced by heat treatment decreased with CTP ampoule treatment (**C**), as visually shown in (**D**); ** p* < 0.05, *** *p* < 0.005, independent samples *t*-test. Scale bar indicates 80 μm. CTP ampoule, Ever Collagen Corrector Collagen Tripeptide Ampoule; AGEs, advanced glycated end products.

**Figure 10 ijms-23-01101-f010:**
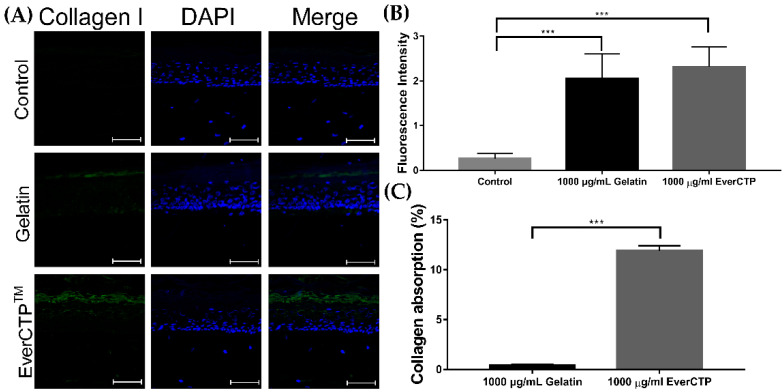
Comparison of collagen 1 expression using immunofluorescence staining (**A**). The 1000 μg/mL gelatin (nonhydrolyzed collagen) treatment collagen induced more collagen 1 expression than the control (**B**). The 1000 μg/mL EverCTP^TM^ treatment induced more collagen 1 expression than the 1000 μg/mL gelatin treatment. Collagen absorption was considerably increased with 1000 μg/mL of EverCTP^TM^ compared to 1000 μg/mL of gelatin (**C**); *** *p* < 0.005, independent samples *t*-test. Scale bar indicates 80 μm.

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
