# Peer review of "Effect of a Topical Collagen Tripeptide on Antiaging and Inhibition of Glycation of the Skin: A Pilot Study"

_ijms, 2022, doi:10.3390/ijms23031101_

Round 1

Reviewer 1 Report

Review of Lee YI et al., Effect of topical collagen tripeptide on antiaging and inhibition of glycation of the skin: A prospective, single-arm clinical trial. Int J Mol Sci 2021.

General Comments on Manuscript
The paper noted above is a comprehensive examination of the influence of a low molecular weight copper tripeptide isolated from Pangasius hypophthalmus skin, a common source of fish-derived collagen.  The paper is well-written, and the grammar and English are good.  The paper includes studies conducted in vivo and in vitro with ample statistical significance that appears to switch between Student t-testing and Wilcoxon sign rank tests.  The switching between statistical analyses techniques leaves the reader suspecting that the statistics applied where the ones that gave authors the results they were seeking.

The references appear to support the discussion and in checking a few of the listed references, the referee noted that they were consistent and accurate.

Review of and Comments on Clinical Studies

The clinical study, done under IRB designation GIRB-21624-DX included 22 women who applied the topical product called “Evercollagen Collagen Corrector Collagen Tripeptide 76” twice daily to the face for 4-weeks. This referee was unable to locate a public reference to this IRB study using Google.  It might be helpful if the authors provide a link to the actual IRB which should be listed in the public domain.  Also, it appears from the web that “Ever Collagen” is two words, not one as the authors have written it in the paper.  

The data accumulated from the clinical study is not compared against a Placebo cream without the copper tripeptide, but rather against the original Baseline measurements.  For this reason, the clinical studies suffer from solid scientific rigor.  This referee tried to determine what additional ingredients are included in the commercial formulation but was unable to determine what these ingredients are that formulate the copper tripeptide.  It might be beneficial if the authors included a summary of the full ingredients available in this product.   It might be a concern, for example, if the serum-like formulation contains ingredients such as glycerin or urea.  In this case, much of the beneficial effects associated with the product may simply be attributable to these more commonly well-known hydrating ingredients.  In addition, images from the web of the serum show it to be very white which implies that the level of copper tripeptide must be quite low as it is well-known that copper tripeptides are a deep blue color (for example, the well-known copper tripeptide Cu-Gly-His-Lys is a very deep blue color).  It would be nice to know how much of the copper tripeptide is contained in the serum even if the level is provided as a range as this is suggested to be the critical active in the formulation.

Review of and Comments on In Vitro Studies

The in vitro studies are well run and well presented.  However, there are some concerns this referee has for some of the data presented.

The authors appear to switch between UVB, H2O2 and Heat to generate various responses in their cell cultures and tissue models.  It is imperative that they include some discussion why each stress was chosen for a particular study. 

The results in Figure 5A are curious.  It appears that hydrolyzed copper peptide used in the studies, which the reader is led to believe is a nutritional version of the same peptide used in the topical product, does not statistically stimulate collagen at the 1000ppm level.  The authors make a note of this inconsistency but don’t address why this happens.  This is important because in Figure 10 the authors demonstrate in their collage absorption models on tissues a very profound influence of the copper tripeptide on collagen synthesis (Figure 10B).  The authors must make a comment on why the results from Fibroblast models were so significantly different than the results from the tissues.

In Figure 6, the authors make a reference to a p-value of 0.00.  This is likely a typo that needs correcting.

The comparison of a very low molecular weight copper tripeptide verses a high molecular weight gelatin as discussed in Figure 10 is somewhat like comparing apples to oranges.  It would be highly anticipated that gelatin would not be able to penetrate the stratum corneum of the KeraSkin Full Thickness tissues.  It would have been a much better comparison for the authors to examine the copper tripeptide against the simple tripeptide that forms the dominant part of the copper tripeptide, namely, Gly-Pro-Hyp.     

Comments on Discussion

The authors make the following statement in the Discussion, in paragraph 5: “The antioxidant properties of the CTP depend on the lower molecular weight of peptides, as smaller molecules with average molecular weights of 5 kDa showed greater ability to donate an electron or hydrogen to stabilize oxygen radicals than larger molecules”, and they make a reference to work by Leon-Lopez et al to support this claim.  However, this statement negates a profound influence in the antioxidant benefits of the copper tripeptide from the copper.  In the work by Leon-Lopez, the studies were done looking at the influence of hydrolyzed sheep skin, not fish skin and, more importantly, not copper-complexed peptides.  It is important that the authors make some comments and references to the role of the copper ions that are bound to the tripeptides as this is, in all actuality, the most likely functional portion of the molecule.  Leon-Lopez used the DPPH assay to demonstrate the antioxidant potential of hydrolyzed sheep skin collagen.  The authors here would benefit from using a similar antioxidant assay such as DPPH or ORAC to demonstrate that the peptides are the antioxidant component and perhaps might be able to show an extended antioxidant benefit from the copper complexing the tripeptide. 

The authors need to expand more deeply their discussion on the role of the copper in the copper tripeptide as far as the functionality of the active is concerned on skin.     

In their summary of the limitations of their study, it is critical for the authors to discuss the deficiencies that are part of a study in which they do not compare the efficacy of their product against a Placebo formulation that does not contain the copper tripeptide. 

Author Response

General Comments on Manuscript
1. The paper noted above is a comprehensive examination of the influence of a low molecular weight copper tripeptide isolated from Pangasius hypophthalmus skin, a common source of fish-derived collagen.  The paper is well-written, and the grammar and English are good.  The paper includes studies conducted in vivo and in vitro with ample statistical significance that appears to switch between Student t-testing and Wilcoxon sign rank tests.  The switching between statistical analyses techniques leaves the reader suspecting that the statistics applied where the ones that gave authors the results they were seeking.

The references appear to support the discussion and in checking a few of the listed references, the referee noted that they were consistent and accurate.

Review of and Comments on Clinical Studies

The clinical study, done under IRB designation GIRB-21624-DX included 22 women who applied the topical product called “Evercollagen Collagen Corrector Collagen Tripeptide 76” twice daily to the face for 4-weeks. This referee was unable to locate a public reference to this IRB study using Google.  It might be helpful if the authors provide a link to the actual IRB which should be listed in the public domain. Also, it appears from the web that “Ever Collagen” is two words, not one as the authors have written it in the paper.  

Answer: Thank you for your kind and thorough review. First of all, the test method for statistical analyses of each parameter were chosen based on the results from the normality test by Kolmogorov–Smirnov test.  Unlike Student's t-test, the Wilcoxon signed-rank test does not assume that the differences between paired samples are normally distributed. Hence, we added the following sentence in the method section for better clarification:

‘Data sets were assessed for normality using the Kolmogorov–Smirnov test. Clinical parameters at baseline and week 4 were compared between groups using the Wilcoxon signed-rank test or paired samples t-test according to the results from the normality test.’ (lines 491-494).

Some of the Korean IRB are not available in public domain. We have attached the English proof form of IRB approval as a supplementary PDF file. Regarding the name for the test product, we reconfirmed with the company and changed the name as “Ever Collagen” in the manuscript.

2. The data accumulated from the clinical study is not compared against a Placebo cream without the copper tripeptide, but rather against the original Baseline measurements.  For this reason, the clinical studies suffer from solid scientific rigor.  This referee tried to determine what additional ingredients are included in the commercial formulation but was unable to determine what these ingredients are that formulate the copper tripeptide.  It might be beneficial if the authors included a summary of the full ingredients available in this product.   It might be a concern, for example, if the serum-like formulation contains ingredients such as glycerin or urea.  In this case, much of the beneficial effects associated with the product may simply be attributable to these more commonly well-known hydrating ingredients.  In addition, images from the web of the serum show it to be very white which implies that the level of copper tripeptide must be quite low as it is well-known that copper tripeptides are a deep blue color (for example, the well-known copper tripeptide Cu-Gly-His-Lys is a very deep blue color).  It would be nice to know how much of the copper tripeptide is contained in the serum even if the level is provided as a range as this is suggested to be the critical active in the formulation.

Answer: Thank you for your valuable comment. We have added the total ingredients of the test ampoule in supplementary table (Table S3, 0.5% EverCTP included as the ingredient). As you mentioned, the ampoule does contain glycerin and various ingredients that could augment the test product’s antiaging effect seen in the clinical study. In order to further investigate and improve evidences for the antiaging effect of the topical collagen tripeptide, we additionally performed in vitro study not only on the serum-formulated test product (named as ‘Ever Collagen Corrector Collagen Tripeptide Ampoule’), but also on the collagen tripeptide raw material itself (EverCTPTM; NEWTREE Co., Ltd., Seoul, Korea). As shown in figure 4 and 5, the in vitro treatment of HDFs with EverCTPTM in its raw material formulation showed increased gene expression and protein levels of collagen 1, while MMP 1, 3, and 9 decreased compared to the UV-exposure group. The treatment with EverCTPTM also resulted in an increased elastase inhibition activity (figure 6) while decreasing ROS (figure 7) and AGEs productions (figure 8), suggesting the possible antiaging effect of EverCTPTM in its raw form.

Regarding copper tripeptide that you mentioned, I believe that the manuscript does not mention “copper” as the raw material, but skin of Pangasius hypophthalmus that contains 4% Gly-Pro-Hyp with hydrolyzed collagen tripeptide content exceeding 25% (lines 346-349). Please let me know if I have misunderstood your inquiry.

Review of and Comments on In Vitro Studies

The in vitro studies are well run and well presented.  However, there are some concerns this referee has for some of the data presented.

3. The authors appear to switch between UVB, H2O2 and Heat to generate various responses in their cell cultures and tissue models. It is imperative that they include some discussion why each stress was chosen for a particular study. 

Answer: Thank you for your comments. We have added the additional information on each stress in Material and Method section as suggested:

‘UVR exposure was used in this experiment for the generation of extrinsic cellular aging on HDFs’ (Lines 132-133)

‘To induce ROS stress as the positive control, the treatment with H2O2 was performed.’ (Line 440-441)

‘To induce denatured collagen production by HDFs as the positive control, heat stimulation was performed.’ (Line 468-469)

4. The results in Figure 5A are curious.  It appears that hydrolyzed copper peptide used in the studies, which the reader is led to believe is a nutritional version of the same peptide used in the topical product, does not statistically stimulate collagen at the 1000ppm level.  The authors make a note of this inconsistency but don’t address why this happens.  This is important because in Figure 10 the authors demonstrate in their collage absorption models on tissues a very profound influence of the copper tripeptide on collagen synthesis (Figure 10B).  The authors must make a comment on why the results from Fibroblast models were so significantly different than the results from the tissues.

Answer: Thank you for your comments. The statistical significance on PIP1 concentration was not reached due to the degree of the standard deviation. We still included the data since the mean values for PIP1 increased from 824.65±8.12 to 885.65±89.69 when treated with 1000ug/ml of EverCTPTM. Additionally, the gene expression results of Collagen 1 as shown in Figure 4A revealed a statistically significant improvement, supporting the possibility that application of test product increased the expression of collagen. In tissues, the ex vivo environment can be considerably different from the cell culture environment. Our group pre-tested 1000 μg/mL of EverCTPTM on the skin tissue and it did not show any tissue toxicity; hence we decided to perform the highest concentration of EverCTPTM that had shown no significant cellular toxicity on the MTT assay (as shown in Figure S1).

5. In Figure 6, the authors make a reference to a p-value of 0.00.  This is likely a typo that needs correcting.

Answer: Thank you for your comment. As you mentioned, we corrected the sentence as shown below:

‘*** p < 0.00’ -> ‘*** p < 0.005’ (Line 178)

6. The comparison of a very low molecular weight copper tripeptide verses a high molecular weight gelatin as discussed in Figure 10 is somewhat like comparing apples to oranges.  It would be highly anticipated that gelatin would not be able to penetrate the stratum corneum of the KeraSkin Full Thickness tissues.  It would have been a much better comparison for the authors to examine the copper tripeptide against the simple tripeptide that forms the dominant part of the copper tripeptide, namely, Gly-Pro-Hyp.

Answer: We sincerely agree with your comments. The ex vivo comparison of collagen absorption between gelatin and the test product was initially performed to visualize that the collagen tripeptide indeed possessed a low molecular weight, hence it could penetrate the skin barrier unlike gelatin. As our objective of the study was to show the effect of ‘topical’ collagen tripeptide on anti-aging, we believed that visualizing skin penetrance of the topical test product was necessary. As mentioned previously, our form of tripeptide did not specifically contain copper; but as you kindly mentioned, a further ex vivo study on comparison of skin penetrance between the test product and the simple tripeptide would be meaningful; therefore, we included the notion in the discussion section as below:

‘Secondly, an additional skin absorption study of collagen 1 between EverCTPTM and the simple tripeptide, Gly-Pro-Hyp, would further show the efficacy and biocompatibility of the test product in its topical formulation.’ (lines 322-325)

Comments on Discussion

7. The authors make the following statement in the Discussion, in paragraph 5: “The antioxidant properties of the CTP depend on the lower molecular weight of peptides, as smaller molecules with average molecular weights of 5 kDa showed greater ability to donate an electron or hydrogen to stabilize oxygen radicals than larger molecules”, and they make a reference to work by Leon-Lopez et al to support this claim.  However, this statement negates a profound influence in the antioxidant benefits of the copper tripeptide from the copper. In the work by Leon-Lopez, the studies were done looking at the influence of hydrolyzed sheep skin, not fish skin and, more importantly, not copper-complexed peptides. It is important that the authors make some comments and references to the role of the copper ions that are bound to the tripeptides as this is, in all actuality, the most likely functional portion of the molecule. Leon-Lopez used the DPPH assay to demonstrate the antioxidant potential of hydrolyzed sheep skin collagen. The authors here would benefit from using a similar antioxidant assay such as DPPH or ORAC to demonstrate that the peptides are the antioxidant component and perhaps might be able to show an extended antioxidant benefit from the copper complexing the tripeptide. 

The authors need to expand more deeply their discussion on the role of the copper in the copper tripeptide as far as the functionality of the active is concerned on skin.     

In their summary of the limitations of their study, it is critical for the authors to discuss the deficiencies that are part of a study in which they do not compare the efficacy of their product against a Placebo formulation that does not contain the copper tripeptide.

Answer: Thank you for your detailed comments. As mentioned previously, our formulation of collagen tripeptide did not specifically include ‘copper.’ If I misunderstood your inquiry, please let me know, and we will try to answer further question thoroughly.

Reviewer 2 Report

This reader is personally experienced with the signs and symptoms of photoageing, and therefore was most interested in the outcomes of the study.  However, I have a problem being convinced by the in vivo data.  The numerical changes are so minute, being around 5%, with errors about an order of magnitude greater than the changes.  These data are not persuasive. In Fig 1E and 1F, 3 subjects’ images are shown, which do not realistically demonstrate the changes that are claimed, and why only 3 subjects shown? On what basis were these selected? Which data are the mean values for all 22 subjects?  This applies also to Fig.2.  In Fig 3, some arrows on the fluorescent images would help identify the changes that are claimed.  However the numerical graphs again show miniscule changes, being difficult to understand how they can be statistically significant. If possible, the authors should investigate better methods to display their in vivo findings, otherwise restrict this paper to the in vitro data, which is stronger and more convincing.

The in vitro data is clearer in Fig. 4.  However the text description lines 132-134 is confusing.

Elastase inhibition % seems a strange choice of expression.  Actual elastase activity, for which there is no evidence here, would seem to be more logical for Fig. 6.

Fig. 8B does not convince of significance; Fig. 8C does not show significance for the UV effect?

In Fig. 10B, the important comparison between gelatin and CTP has not been made.

Lines 273, 278 refer to ‘migration’ of the dermal fibroblasts.  These cells do not naturally migrate within the dermis, therefore this reference needs explanation.

Author Response

1. This reader is personally experienced with the signs and symptoms of photoageing, and therefore was most interested in the outcomes of the study. However, I have a problem being convinced by the in vivo data. The numerical changes are so minute, being around 5%, with errors about an order of magnitude greater than the changes. These data are not persuasive. In Fig 1E and 1F, 3 subjects’ images are shown, which do not realistically demonstrate the changes that are claimed, and why only 3 subjects shown? On what basis were these selected? Which data are the mean values for all 22 subjects?  This applies also to Fig.2. In Fig 3, some arrows on the fluorescent images would help identify the changes that are claimed. However, the numerical graphs again show miniscule changes, being difficult to understand how they can be statistically significant. If possible, the authors should investigate better methods to display their in vivo findings, otherwise restrict this paper to the in vitro data, which is stronger and more convincing.

Answer: Thank you for your thorough comments. Our investigators sincerely agree that 4-weeks use of topical antiaging product such as collagen tripeptide used in our study would not sufficiently show enough clinical changes. Although we performed extensive laboratory studies to confirm the antiaging properties of the topical collagen tripeptide, a further prospective, controlled study is definitely needed to support the results from this pilot study. Hence, we modified the title as ‘Effect of a topical collagen tripeptide on antiaging and inhibition of glycation of the skin: A pilot study’ and added to the discussion section as the following:

‘Hence, a further clinical study with a larger sample size and longer follow-up period with a double-blinded, controlled design is required.’ (lines 321-322).

The percent improvement, mean values, and standard deviations from baseline and week 4 for all subjects are additionally listed on Table S1 as kindly suggested. The selected subjects are from those who gave permission to use their digital images that are cropped to be unidentifiable. We agree that PRIMOS and Antera 3D images shown in Figure 1E, F, and Figure 2B does not sufficiently demonstrate improvements in reduction of wrinkles; hence we tried to insert arrows on wrinkles from baseline and week 4 to better display the reduction in the number of visible wrinkles. Also, we added arrows on the ultrascan image to show where the device measured.

Lastly, we added the patient satisfaction score we performed at the end of the study to back up the efficacy measurement data as the following:

‘Overall, the average percentage of subjects who responded to “very satisfied” and “satisfied” to the use of topical CTP product and the overall improvement in facial skin problems caused by aging was 95.5% (“very satisfied”: 4 subjects, “satisfied”: 17 subjects). One subject rated “slightly satisfied” to the test product and the overall improvement of the facial skin. No adverse events were observed during the 4-week study period, and none of the participants dropped out of the study because of adverse events, suggesting that the test product formulation was safe to use.’ (lines 110-116).

2. The in vitro data is clearer in Fig. 4. However, the text description lines 132-134 is confusing.

Answer: Thank you for your comment. We changed the manuscript as the following:

‘The expression levels of MMP-1, -3, and -9 reduced significantly after the treatment with 250 μg/mL, 500 μg/mL, and 1000 μg/mL of EverCTPTM compared to the UVR exposure group.’ (lines 137-139).

3. Elastase inhibition % seems a strange choice of expression. Actual elastase activity, for which there is no evidence here, would seem to be more logical for Fig. 6.

Answer: Thank you for your comment. The elastase inhibition assay was performed as recommended by Korean Ministry of Food and Drug Safety. Previous studies which performed the identical ‘elastase inhibition assay’ is as listed in the following: 1) Figure 2B; Park SH, Yi YS, Kim MY, Cho JY. Antioxidative and Antimelanogenesis Effect of Momordica charantia Methanol Extract. Evid Based Complement Alternat Med. 2019 May 2;2019. 2)  Figure 7; Tsuji N, Moriwaki S, Suzuki Y, Takema Y, Imokawa G. The role of elastases secreted by fibroblasts in wrinkle formation: implication through selective inhibition of elastase activity. Photochem Photobiol. 2001 Aug;74(2):283-90.

Thus, to clarify the method, we modified the method section as shown below:

‘To measure elastase inhibition activity, an elastase solution extracted by HDF cells was prepared by freezing/thawing the cells more than three times. After the solution was centrifuged (3000 rpm) at 4°C for 20 minutes, the supernatant was harvested and used as the elastase solution. The elastase solution was placed in 100 μg of protein per well on a 96-well plate, and we added 0.2-M Tris-HCl buffer (pH 8.0) to make the total volume 88 μL. We added 2 μL of N-succinyl-tri-alanyl-p-nitroanilide, a substrate of elastase, and 10 μL of EverCTPTM of three concentrations or 10 μL of 1-mM PR (Sigma Aldrich) to each well containing the elastase solution and cultured the plate at 37°C for 90 minutes. We added 10 μL 0.2-M Tris-HCl buffer as a control group. Finally, using an ELISA microplate reader (VersaMax), the absorbance was measured at 405 nm at 37°C. The elastase inhibition activity was expressed as a decrease (%) in absorbance of the test group with and without the sample solution.

Elastase inhibition activity (%) = (absorbance of control – absorbance of the test sample) / absorbance of control x 100’ (lines 420-434)

4. Fig. 8B does not convince of significance; Fig. 8C does not show significance for the UV effect?

Answer: Thank you for your valuable comments. We have double-checked the statistical significance on 8B using raw data from three independent experiments, and resulted in the p values as shown in the figure. We have included each mean value and SDs in the manuscript as shown below:

‘Treatment with 1000 μg/mL of EverCTPTM significantly decreased the productions of pentosidine (from 211.682 ± 0.425 to 156.975 ± 0.135), methylglyoxal (from 5.586 ± 0.014 to 5.307 ± 0.001) and AGEs production (from 16.688 ± 0.235 to 10.762 ± 0.424), while significantly increasing the production of glyoxalase I compared to the UV group (from 0.008 ± 0.001 to 0.777 ± 0.003; Figure 8, p < 0.005).’ (lines 186-191).

Regarding 8C, we did not include a p value for UV effect since the p value did not meet statistical significance (p = 0.4439). Although the mean value of glyoxalase I activity of the UV exposure group was notably lower than that of the control group, the significance was not reached due to the degree of SD of the control group from three independently repeated experiments. We still included the data as the treatment with 1000 μg/mL of EverCTPTM significantly increased the activity of glyoxalase I.

5. In Fig. 10B, the important comparison between gelatin and CTP has not been made.

Answer: Thank you for your comment. We performed immunofluorescence staining as well as high-performance liquid chromatography (HPLC) in order to not only visualize but also to quantitatively analyze and compare the rates of collagen absorption between gelatin treatment group and EverCTPTM treatment group. Although the florescence intensities measured from the immunofluorescence staining on ex vivo skin did not significantly differ between the gelatin group (2.051 ± 0.550) and the EverCTPTM treatment group (2.316 ± 0.444), the quantitative analysis of collagen absorption rate via HPLC differed significantly (0.425 ± 0.100 and 12.221 ± 0.684, p<0.005). We modified the manuscript for better understanding as shown below:

‘The fluorescence intensity of collagen 1 absorbed into the skin after 1000 μg/mL of EverCTPTM treatment was 2.316 ± 0.444, which was increased compared to those after 1000 μg/mL of gelatin treatment (2.051 ± 0.550) and the negative control (0.266 ± 0.111; Figure 10A–B, p < 0.005). Although shown an increase in EverCTPTM treatment group, the difference in florescent intensities compared to the gelatin group did not reach a statistical significance (p = 0.5520). 

Meanwhile, the results of the additional quantitative evaluation of the rates of collagen absorption via high-performance liquid chromatography (HPLC) compared between the gelatin and EverCTPTM treatments are shown in Figure 10C. The rates of collagen absorption in the gelatin treatment group and EverCTPTM treatment group were 0.425 ± 0.100 and 12.221 ± 0.684, respectively, thus showing a significantly improved quantitative absorption rate of collagen in the EverCTPTM group (p < 0.005).’ (lines 231-242)

6. Lines 273, 278 refer to ‘migration’ of the dermal fibroblasts. These cells do not naturally migrate within the dermis, therefore this reference needs explanation.

Answer: Thank you for your valuable comment. ‘Migration’ of fibroblasts is the term used for the in vitro wound healing assay as referred previously. For instance, reference 18 revealed the impact of glyoxal exposure on cell behavior and particularly on cell proliferation and migration. The authors investigated the migration of the fibroblasts in the presence and absence of glyoxal using an in vitro monolayer wound healing assay; as the result, compared to the rapid wound healing in the controls, the glyoxal treated cells showed delayed wound healing, thus slower migration of the fibroblasts (Guillon C, Ferraro S, Clément S, Bouschbacher M, Sigaudo-Roussel D, Bonod C. Glycation by glyoxal leads to profound changes in the behavior of dermal fibroblasts. BMJ Open Diabetes Res Care. 2021 Apr;9(1)).

Round 2

Reviewer 1 Report

The updated paper appears to address the major issues that were noted in the earlier reading.  It appears to be a more acceptable paper at this time

Reviewer 2 Report

The reviewer's comments have each been addressed and the manuscript is therefore improved.